# ER and PI3K Pathway Activity in Primary ER Positive Breast Cancer Is Associated with Progression-Free Survival of Metastatic Patients under First-Line Tamoxifen

**DOI:** 10.3390/cancers12040802

**Published:** 2020-03-27

**Authors:** Anieta M. Sieuwerts, Márcia A. Inda, Marcel Smid, Henk van Ooijen, Anja van de Stolpe, John W. M. Martens, Wim F. J. Verhaegh

**Affiliations:** 1Department of Medical Oncology and Cancer Genomics Netherlands, Erasmus MC Cancer Institute, Erasmus MC, Doctor Molewaterplein 40, 3015 GD Rotterdam, The Netherlands; 2Philips Research, Precision Diagnostics Department, High Tech Campus 11, 5656 AE Eindhoven, The Netherlands

**Keywords:** signaling pathway, functional activity, metastasis, progression-free survival, tamoxifen, treatment response, mRNA expression, Bayesian network

## Abstract

Estrogen receptor positive (ER+) breast cancer patients are eligible for hormonal treatment, but only around half respond. A test with higher specificity for prediction of endocrine therapy response is needed to avoid hormonal overtreatment and to enable selection of alternative treatments. A novel testing method was reported before that enables measurement of functional signal transduction pathway activity in individual cancer tissue samples, using mRNA levels of target genes of the respective pathway-specific transcription factor. Using this method, 130 primary breast cancer samples were analyzed from non-metastatic ER+ patients, treated with surgery without adjuvant hormonal therapy, who subsequently developed metastatic disease that was treated with first-line tamoxifen. Quantitative activity levels were measured of androgen and estrogen receptor (AR and ER), PI3K-FOXO, Hedgehog (HH), NFκB, TGFβ, and Wnt pathways. Based on samples with known pathway activity, thresholds were set to distinguish low from high activity. Subsequently, pathway activity levels were correlated with the tamoxifen treatment response and progression-free survival. High ER pathway activity was measured in 41% of the primary tumors and was associated with longer time to progression (PFS) of metastases during first-line tamoxifen treatment. In contrast, high PI3K, HH, and androgen receptor pathway activity was associated with shorter PFS, and high PI3K and TGFβ pathway activity with worse treatment response. Potential clinical utility of assessment of ER pathway activity lies in predicting response to hormonal therapy, while activity of PI3K, HH, TGFβ, and AR pathways may indicate failure to respond, but also opens new avenues for alternative or complementary targeted treatments.

## 1. Introduction

Estrogen receptor (ER) positive breast cancer patients are treated routinely with surgical tumor resection, local radiation, and chemotherapy depending on stage and tumor subtype, followed by adjuvant hormonal therapy, either tamoxifen or an aromatase-inhibitor, for at least 5 years [1]. Unfortunately, in several patients, metastases occur either during or following adjuvant treatment, raising the question whether endocrine adjuvant therapy for these patients was the optimal treatment. Choice of endocrine treatment assumes that the ER signal transduction pathway is in the active state and represents the tumor-driving pathway, but this assumption may be questioned in these patients. To that end, Bayesian network-based computational models have been developed and validated biologically to assess the functional activity of signal transduction pathways in a cancer tissue sample in a quantitative manner, based on gene expression levels of pathway-specific target genes [2]. Initial results of the ER pathway model provide evidence that the ER pathway is not always active in breast cancer tissue from patients in which routine ER evaluation indicates ER positivity, and also that ER pathway activity as identified by the model is associated with improved clinical outcome after adjuvant hormonal treatment [2]. The pathway model series has recently been extended to include, besides Wnt and ER, also androgen receptor (AR), PI3K-FOXO, Hedgehog (HH), NFκB and TGFβ pathway analysis models [3,4].

To investigate the clinical utility of these signaling pathway models in terms of predicting therapy benefit, and of the ER pathway model in particular, we performed a multiple pathway model analysis on Affymetrix mRNA expression data from primary tumor tissue samples of a retrospective series of patients with ER positive breast cancer who did not receive adjuvant hormonal treatment and subsequently developed metastatic disease. Upon diagnosis of metastatic disease, these patients were treated with first-line endocrine therapy (tamoxifen). The primary purpose of this analysis was to investigate whether ER pathway activity in the primary tumor is a better predictor of response to tamoxifen or disease progression in this setting than mere ER protein expression. The second purpose was to investigate whether activity of specific signaling pathways is associated with a worse clinical outcome following start of endocrine therapy.

## 2. Materials and Methods

### 2.1. Patient Population

The protocol to study biological markers associated with disease outcome in the current cohort was approved by the medical ethics committee of the Erasmus Medical Center Rotterdam, The Netherlands (MEC 02.953) and tissues were studied in accordance with the Code of Conduct of the Federation of Medical Scientific Societies in the Netherlands (http://www.federa.org/codes-conduct).

The investigated historic cohort from the late 1990s consists of 132 patients who had primary ER positive breast cancer, did not receive any adjuvant hormonal therapy, and developed metastatic disease treated with first-line tamoxifen. Detailed clinical follow-up information was available. Tumor content was determined on fresh frozen tissue samples as described before [5]. Tissue samples from the primary tumor contained at least 30% invasive tumor cells uniformly distributed over 70% of the tissue area. Tissue samples from metastases were unfortunately not available.

RNA was isolated from the fresh frozen primary tumor tissue samples as previously described [5] and had an RNA integrity number (RIN) of 7 or higher. Affymetrix microarray expression analysis was performed (see below). Two samples were removed because they failed microarray quality control (QC) [4], leaving a total of 130 sample microarray data for further analysis. Clinical outcome data used in the analysis were progression-free survival (PFS) and therapy response. Since official RECIST criteria did not exist at the time this cohort was collected, therapy response was determined according to radiological examination by standard Unio Internationale Contra Cancrum (UICC) criteria [6], with tumor size being determined by the longest diameter. An overview of patient characteristics is shown in Table 1. Per patient info is available in Appendix A.

Objective response was observed in 18 patients with metastatic disease, of which 3 had a complete remission (CR), and 15 had a partial remission (PR). There were 33 patients with progressive disease (PD), with an increase in tumor size of 25% or more or showing new tumor lesions within 3 months. The remaining 79 patients were considered to be patients with stable disease (SD), of which 69 patients had SD at > 6 months and 10 patients had SD ≤ 6 months. The median PFS times of the different response groups were: CR, 32 months; PR, 20 months; SD > 6 months, 14 months; SD ≤ 6 months, 5 months; and PD, 3 months. Because the patients with SD > 6 months had a PFS similar to patients with partial remission, we classified them together with CR and PR as responders to tamoxifen as advised by the European Organization for Research and Treatment of Cancer [8].

### 2.2. ER, PR, ERBB2, and PIK3CA Testing

Routine ER and PR protein expression analysis was performed on this historical cohort on a tissue sample of the primary tumor by ligand binding assay or enzyme immunoassay as described before [9]. The cut-off point used to classify tumors as ER or PR positive was 10 fmol/mg cytosolic protein. RNA expression of *ESR1*, *PGR* and *ERBB2* was performed using RT-qPCR as described before [5,10]. *PIK3CA* mutation status (covering E542K, E545A/G/K, H1047L/R) was assessed by SnaPshot multiplex analyses as described in [11].

### 2.3. Microarray Analysis

RNA was isolated from 30 μm sections of a fresh frozen primary tissue sample with RNA-B (Tel Test, Thermo Fisher Scientific Inc., Waltham, MA, USA) and DNAse treated as described before [12]. Affymetrix HG-U133+PM microarray analysis was performed by ServiceXS (Leiden, The Netherlands), and data was provided to Philips Research for subsequent pathway analysis in a blinded manner (data is available from Gene Expression Omnibus under accession number GSE82172). Raw Affymetrix CEL files from the study samples, measured on Affymetrix HG-U133+PM arrays, as well as from pathway calibration samples measured on Affymetrix HG-U133Plus2.0 arrays, were processed using fRMA [13] with ‘random effect’, using fRMA vectors from the Plus2.0 arrays, while only probes were used that are available on both array types, to make processed data comparable.

### 2.4. Signaling Pathway Model Interpretation

Functional signal transduction pathway activity levels were assessed by means of measuring and interpreting pathway-specific target gene expression levels. To this end, Bayesian network models were developed as originally presented for the Wnt and ER signaling pathways in [2]. This approach was subsequently repeated for the AR, PI3K-FOXO, Hedgehog (HH), NFκB and TGFβ pathways as described in [3,4]. In brief, such a Bayesian network, as shown in Figure 1, represents the transcriptional program of a pathway by means of three layers of computational nodes: (a) a transcription factor complex (TC) executing the function of the respective signaling pathway, (b) well validated target genes of the TC and (c) measurement nodes of the target genes’ expression levels. For the measurement nodes, probeset intensities were taken from fRMA processed Affymetrix HG-U133Plus2.0 or Affymetrix HG-U133+PM microarrays as described above. Each pathway model describes (i) how expression of target genes depends on transcription complex activation, and (ii) how gene expression measurements in turn depend on expression of the target genes. Probabilistic relations (i) between the transcription complex and target gene nodes were manually set to ensure good generalization behavior across tissue types. Probabilistic relations (ii) between target genes and their measurement nodes have been calibrated on microarray data of either human cell line experiments or patient samples that are known to have a functionally active or inactive pathway, as described before [2,3,4]. For use in the current study, for all seven models, calibration data were used from Affymetrix HG-U133Plus2.0 arrays, but processed as described above to be compatible with the HG-U133+PM measurements of the study samples.

Once built, the models were biologically validated on human cell line models for the respective signal transduction pathways and on tissue samples of cancer types that are well characterized with respect to activity of the specific signal transduction pathway [2,3,4]. These concerned controlled experiments on e.g., MCF7, BT20, BT474, MDA-MB-231, MDA-MB-453, HCC827, A549 and LNCaP cell lines and isolated T-lymphocytes and macrophages, in which compounds were added to stimulate or inhibit pathway activation. Furthermore, well characterized tissue samples were used, such as normal colon vs. colon adenomas, basal cell carcinomas (BCC) vs. normal skin and medulloblastomas with known sonic hedgehog (SHH) activity or beta-catenin mutation vs. other. For more details, we refer to [2,3,4].

After calibration, the computational signaling pathway models were frozen, and used to interpret microarray data of tumor samples included in this study. For each individual sample, the target genes’ expression measurements (c) were entered in each pathway model and then the model inferred backwards what the odds were that the respective TC must have been active vs. inactive. These odds are generally represented on a logarithmic scale (log2odds). Log2odds larger than zero (i.e., odds > 1) indicate that there is more evidence for an active TC, and log2odds smaller than zero that there is more evidence for an inactive TC, which corresponded well with the ‘ground truth’ status with respect to signaling pathway activity of calibration and biological validation samples. Therefore, we use a threshold of 0 on the log2odds scale in this paper to separate low from high pathway activity.

For the pathway models, calculated log2odds scores represent activity of their respective TCs, which for most pathways is directly related to activity of the pathway. However, there is an inverse relation between PI3K pathway activity and FOXO activity, as the FOXO transcription factor is generally inactivated by an active PI3K pathway. Hence, an inactive FOXO transcription factor is indicative of an active PI3K pathway. Alternatively, FOXO activity can be induced by oxidative stress, indicated by a high *SOD2* expression level, which is with high likelihood also associated with an active PI3K pathway as described in [4]. To this end, thresholds have been determined on the *SOD2* probesets similarly as in [4], but with the microarray processing done as described above.

### 2.5. Statistics

Univariate and multivariate Cox proportional hazards regression was used to associate the pathways’ activities with PFS on first-line tamoxifen treatment, without and with traditional response prediction factors. This was complemented by a Kaplan–Meier analysis of PFS. Fisher exact tests were used to analyze pathway co-activity, and significant patterns of pathway activity across response categories. All analyses were performed in R [14].

## 3. Results

### 3.1. Pathway Activity

All patients in the cohort were assessed as ER positive luminal breast cancer patients based on conventional pathology diagnostics, but we have shown before that this does not mean that they all had a high ER pathway activity, and that in tumors with low ER activity another signal transduction pathway may be more likely to drive tumor growth [2]. This may have clinical relevance for hormonal treatment efficacy in both the primary and the metastatic tumors of these patients. Therefore we performed, in addition to assessment of ER pathway activity, a broad analysis of signaling pathway activity in the primary tumor samples and related this to metastasis treatment outcome of the patients included in this study.

An overview of the resulting pathway activity scores, in terms of log2odds, is given in Figure 2A. Even though all tumors were ER positive, the ER pathway was measured high in 53 (41%) of the samples using the default threshold on the log2odds scale of 0. With no significant correlation between ER activity (low vs. high) and ER protein levels (Spearman rho = −0.075, *p* = 0.40) and neither between ER activity and *ESR1* RT-qPCR levels (Spearman rho = 0.06, *p* = 0.51), these results confirm that ER expression did not necessarily imply ER transcription factor activation.

The PI3K pathway was found high in 10 (8%) cases, as inferred from low FOXO transcription factor activity, while in 25 patients (19%), high FOXO activity was associated with elevated *SOD2* expression, indicating cellular oxidative stress as an alternative cause of FOXO activity [4,15]. Although in the latter tumors, PI3K pathway activity cannot be formally inferred from FOXO activity, these tumors are generally assumed to be also PI3K pathway active [4,16]. Adding these cases to the FOXO-low-PI3K-high tumors, a total of 35 (27%) tumors were considered to have high PI3K pathway activity. Like the ER pathway, total PI3K activity was not significantly correlated with HER2/*ERBB2* status (Fisher’s test *p* = 0.14, OR = 2.3, 95% CI: 0.66–7.81), indicating that PI3K activity cannot be inferred from HER2/*ERBB2* amplification status. It was also not significantly correlated with *PIK3CA* mutation status (Fisher’s test *p* = 1, OR = 1.0, 95% CI: 0.26–3.78).

Of the other measured signaling pathways, the NFκB pathway was most often scored high, in 74 (57%) cases, and the ER pathway second most with 53 (41%) cases. The other pathways had a high activity in smaller numbers of tumors (Figure 2A). In 108 (83%) of the 130 samples, at least one of the seven measured pathways was found high (at a log2odds threshold of 0); see Figure 2B. In 59 samples (45%), two or more pathways were highly active. 

### 3.2. Pathway Activity Combinations

Signaling pathways not only exert cellular functions independently, but also in concert. To obtain more insight into disease pathophysiology it is useful to investigate combinations of signaling pathway activities. 

Similar to the FOXO-PI3K pathway, the NFκB pathway plays a role in protecting cells from oxidative stress caused by reactive oxygen species (ROS) [17]. For the FOXO-PI3K pathway, this condition can be recognized by elevated levels of the superoxide dismutase (*SOD2*) gene, which codes for a protein that protects against oxidative damage [4]. In 24 out of the 25 FOXO-high-*SOD2*-high PI3K active samples (96%), categorized as a cellular state of oxidative stress, the NFκB pathway also had a high activity (OR = 25.9, *p* = 3.0 × 10^−6^; see Figure 3, NFκB panel, top row) confirming the oxidative stress condition. Conversely, when FOXO was measured low, indicating high PI3K pathway activity without cellular oxidative stress, the NFκB pathway was nearly always low (9 out of 10 FOXO-low PI3K active samples had low NFκB activity, OR = 13.7, *p* = 0.002). 

Next, we analyzed which signaling pathways were preferentially activated in the three subgroups of our ER positive patient cohort as defined by the combinations of the PI3K-FOXO pathway and *SOD2* gene expression, being: (1) PI3K pathway activity low, as defined by high FOXO activity and low *SOD2*; (2) PI3K pathway activity high, defined by low FOXO activity; (3) putative high PI3K pathway activity with cellular oxidative stress, defined by high FOXO activity and high *SOD2*, also with high NFκB pathway activity (Figure 3, top row). AR, ER, and HH pathway activity were not specifically linked to any of the groups, but high TGFβ pathway activity was most frequently present in the PI3K pathway-oxidative stress subgroup, and therefore most frequently associated with high NFκB pathway activity (12 of the 14 high TGFβ pathway activity samples also had high NFκB pathway activity, OR = 5.2, *p* = 0.02). Furthermore, high TGFβ pathway activity was also associated with high HH pathway activity (4 out of the 7 high HH pathway activity samples also had high TGFβ pathway activity, OR = 15.1, *p* = 0.003; Figure 3, bottom row). Samples with high HH pathway activity all had high NFκB pathway activity (OR = ∞, *p* = 0.019) and low ER pathway activity (OR = 0, *p* = 0.041).

On the other hand, in the low FOXO activity (high PI3K pathway activity) subgroup, hardly any other high pathway activity was found (in one sample the NFκB pathway was active, in three other samples the ER pathway was active), indicating that in general in this subgroup the PI3K pathway was the dominant identified driving pathway.

### 3.3. Association of Signaling Pathway Activity with Progression-Free Survival

With respect to PFS of patients with metastatic disease, univariate Cox proportional hazards regression of the signaling pathways being high or low (log2odds threshold of 0) showed that high ER pathway activity in the primary tumor was associated with a longer time to progression of metastases, with a hazard ratio of 0.58 (*p* = 0.005), and that high PI3K pathway activity in the primary tumor was associated with a shorter time to progression, with a hazard ratio of 1.8 (*p* = 0.004); see Figure 4A. The latter did not depend on the kind of PI3K activity, as the low FOXO/high PI3K activity group and the oxidative stress-associated PI3K activity group had comparable hazard ratios of 1.86 and 1.81, respectively, when compared to the low PI3K activity group. High AR and HH pathway activity were also associated with a shorter PFS, with a hazard ratio of 4.2 and 2.2, respectively (*p* = 0.05 and 0.051, respectively). Notice that only two patients were classified as having high AR pathway activity, in combination with low ER and PI3K pathway activity. All seven patients with high HH pathway activity also had low ER pathway activity. A multivariate analysis of these four pathways showed similar results, although the p-value for the HH pathway increased to 0.12 (Figure 4B).

Figure 4E–H show Kaplan–Meier plots for ER, PI3K, AR, and HH pathway activity. Median PFS increased from 8.7 months for ER pathway inactive patients to 12.4 months for ER pathway active patients (logrank *p* = 0.004), and decreased from 12.4 months for PI3K pathway inactive patients to 5.2 months for PI3K pathway active patients (logrank *p* = 0.003). Median PFS decreased from 10.1 months for AR inactive to 4.0 months for the two AR pathway active patients (logrank *p* = 0.033), and it decreased from 10.6 months for HH pathway inactive patients to 6.6 months for HH pathway active patients.

### 3.4. Association of ER and PI3K Pathway Activity with Progression-Free Survival in Combination with Clinical Predictors

To exclude the possibility that the correlation between ER or PI3K pathway activity and PFS was due to a dependency on other (known) predictors for PFS, a multivariate analysis was performed for PFS with ER and PI3K pathway activity in combination with known predictive factors: disease-free interval (DFI), dominant site of relapse, menopausal status at start 1st line therapy, log *ESR1* and *PGR* RT-qPCR expression, HER2/*ERBB2* RT-qPCR status and *PIK3CA* mutation status; see Figure 4. Univariate analysis for PFS indicated that DFI and HER2/*ERBB2* status are also significant prognostic factors for PFS in this cohort (HR =0.47 for DFI > 3yr and 1.85 for amplified *ERBB2*, with *p* = 0.0076 and 0.027, respectively); see Figure 4C. In a multivariate analysis using only the significant predictors from the univariate analysis, DFI and activity of ER and PI3K pathways remained significant (HR = 0.54 for DFI > 3yr, 0.64 for ER and 1.7 for PI3K; *p* = 0.038, 0.035 and 0.015, respectively), while HER2/*ERBB2* status did not (HR = 1.43, p = 0.21); see Figure 4D, top part. Without HER2/*ERBB2* status, which was unknown for ten patients, the results were even slightly better (Figure 4D, bottom part). Hence, ER and PI3K pathway activity appear to be independent predictive markers for PFS. The Kaplan–Meier plots (Figure 4E,F) already illustrated the favorable and unfavorable effect of high ER and PI3K pathway activity, respectively, on PFS. 

### 3.5. Association of Specific Signaling Pathway Activity with Response

Finally, we assessed whether pathway activity in this study was associated with the clinical response categories. While ER pathway activity was clearly associated with a longer PFS, there was no significant relation with the five response categories (CR, PR, SD > 6 m, SD ≤ 6 m, PD; Fisher test p = 0.36; see Table 2A). The PD group seemed to have fewer patient samples with high ER activity, but this was not significant.

PI3K pathway activity, however, did show an association with the five response categories (Fisher test p = 0.035), showing fewer patient samples with high PI3K pathway activity in the response group (CR, PR, SD > 6 m) than in the non-response group (OR = 3.5, p = 0.003); see Table 2B. In addition, TGFβ pathway activity was more often high in patients with PD than non-PD (OR = 4.8, p = 0.008); see Table 2C.

## 4. Discussion

In this study, we analyzed whether the molecular phenotype of a cancer is predictive for therapy response. To this end, we used novel computational models that measure functional activity of oncogenic signaling pathways, which have been developed over the past decade for activity assessment of the AR, ER, PI3K-FOXO, HH, NFκB, TGFβ and Wnt pathways based on analysis of mRNA from cancer tissue [2,3,4]. In the clinical study presented here, these pathway models were used to analyze signaling pathway activity in patient tissue samples from a historic cohort of primary ER positive breast cancer patients who were hormonal treatment-naïve until they developed metastatic disease and were treated with tamoxifen in the first-line. Since only patients were selected who were ER+ and who developed metastases, variation in tumor aggressiveness may be assumed to have been relatively small, which reduced the risk of confounding factors in the analysis.

### 4.1. The ER Pathway

In this study, of 130 ER positive patients, the ER pathway model identified a subgroup of 53 (41%) patients in which functional ER pathway activity was measured as high, using a preset threshold of 0 on the log2odds score [2]. This fraction is slightly lower than earlier observations and likely due to the bias in the study from selecting only patients who developed metastatic disease, since ER pathway activity is known to be associated with a better prognosis [18]. The observation that only a subset of ER positive tumors actually have high ER pathway activity, and that the levels of ER expression and ER pathway activity are poorly correlated, are in full agreement with our previous reports showing that ER expression is a prerequisite, but not sufficient, for pathway activity [2,19]. Indeed, the ER protein is not necessarily in its active transcribing form, and does not always indicate an active ER pathway [2,20]. The current results confirm that the ER protein expression level is not a sufficiently specific biomarker for functional ER pathway activity.

### 4.2. The PI3K Pathway

Measurement of HER2/*ERBB2* expression is used in routine diagnostics to obtain information on activity of the HER2-induced PI3K pathway, and more importantly to predict response to HER2 targeting drugs such as trastuzumab [21]. In the current study, no significant correlation was found between HER2/*ERBB2* mRNA expression status as measured by RT-qPCR and activity of the downstream PI3K pathway, suggesting that HER2 overexpression alone does not necessarily indicate PI3K pathway activity. This is in agreement with the emerging consensus that the predictive value of HER2 protein expression for response to PI3K pathway targeting drugs has been disappointing, and the biomarker seems to lack both sensitivity and specificity [22]. A possible explanation for this is the complex regulation of PI3K-AKT pathway activity that also occurs downstream or independent of the HER2/Neu membrane receptors [4,22]. Also, *PIK3CA* mutation status was not significantly correlated with PI3K activity. Like HER2 overexpression, a mutation upstream in a signaling pathway does not necessarily lead to activation downstream.

### 4.3. Signal Transduction Pathway Combinations

Signal transduction pathways can interact to regulate cellular processes such as cell division, differentiation, and migration. We described previously that the well-known inverse relationship between FOXO transcription factor activity and PI3K pathway activity can be broken by the presence of cellular oxidative stress, which activates the FOXO transcription factor irrespective of PI3K pathway activity. However, the presence of cellular oxidative stress itself probably reflects growth factor pathway activity, specifically activity of the PI3K pathway [4,23]. Under such circumstances, FOXO becomes transcriptionally active to protect the cell against ROS by transcription of specific genes such as *SOD2*. The NFκB pathway is similarly activated in the presence of oxidative stress [24,25,26]. Thus, combined activity of both FOXO and NFκB pathways, in the presence of high *SOD2* expression, is strongly indicative of cellular oxidative stress [27,28]. Such a cellular stress condition with protection against DNA damaging effects, thereby preventing apoptosis, has been reported to reflect aggressive cancer with high proliferation rates and risk at metastasis [4,29,30]. Based on this connection, the samples were divided into three subgroups: (1) low PI3K pathway activity, as defined by high FOXO activity and low *SOD2*; (2) high PI3K pathway activity, defined by low FOXO activity; (3) putative PI3K pathway activity with cellular oxidative stress, defined by high FOXO activity and high *SOD2*, in the presence of high NFκB pathway activity. In the oxidative stress group, 17 (68%) patients had low ER pathway activity, five of which with high TGFβ and two with high HH pathway activity, suggesting a tumor subtype with some characteristics of epithelial mesenchymal transition [31,32,33]. On the other hand, in the PI3K pathway active subgroup without oxidative stress, the PI3K pathway seemed to be mostly the single tumor-driving pathway.

Comparing these three PI3K pathway activity groups to the pathway activity profiles found in other sets of breast cancer tumors [34], group 1 with low PI3K pathway activity seems to most closely resemble the luminal A breast cancer subtype with nearly half of the samples having high ER pathway activity, low PI3K and low TGFβ pathway activity. The third group (putative PI3K pathway activity with cellular oxidative stress), despite eight (32%) samples with a high ER pathway activity, shows distinct features that have been described in triple-negative tumors, i.e., TGFβ activity and oxidative stress [4,29,35]. TGFβ is thought to exert its tumor promoting effects through induction of EMT in cancer cells, a process which may be promoted by cellular oxidative stress, and associated with aggressive tumor pathology and bad prognosis [4,29,30,31,33,36,37]. 

Other potentially relevant signaling pathway combinations were found in the seven samples with high HH pathway activity, which all had high NFκB and low ER pathway activity. The HH/NFκB pathway activity combination may reflect the synergistic crosstalk between these two pathways [38]. Another interesting combination was NFκB pathway activity combined with FOXO activity but with low *SOD2* levels, thus without the protective effect of *SOD2*. This may be indicative of apoptosis [17,39,40], or the presence of immune infiltrate.

### 4.4. Primary Tumor Pathway Activity to Predict Response to Tamoxifen in Metastases and Progression-Free Survival

The pathway models have been developed with the main goal of predicting response to targeted therapy, e.g., hormonal therapy. The fact that patients in this cohort had not received any endocrine treatment prior to the tamoxifen treatment of metastatic disease may have improved the predictive value of ER pathway activity in the primary tumor for the PFS, since no secondary endocrine treatment resistance mechanisms were in play [41]. Persistence of ER pathway activity in metastases of several ER active primary tumors provides a logical explanation for the longer PFS in these patients. Furthermore, since tamoxifen is known to be a partial agonist of the ER transcription factor in the absence of estradiol, it may stimulate growth of ER positive tumors that have an inactive pathway due to lack of endogenous estradiol. Hence, a treatment-naïve inactive ER pathway in ER positive metastases may explain a shorter PFS under tamoxifen treatment, contributing also to the predictive value of ER pathway activity measured in the primary tumor. 

On the other hand, the predictive value of primary tumor ER pathway activity may have been limited by several factors, among which differences between the primary tumor and metastases. For instance, an active ER pathway in the primary tumor need not imply activity in the metastases, potentially resulting in disease progression, possibly worsened by the above-mentioned agonistic effect of tamoxifen. The other way around may also occur, as one patient had low ER activity in the primary tumor, while she had a complete response on first-line tamoxifen treatment of the metastasis. Heterogeneity in ER staining between primary tumor and metastases is known to occur frequently, as well as between various metastases of the same patient, suggesting variations in ER activity [42,43,44,45,46,47]. In a recent study in which ER pathway activity was compared between primary and metastatic tumors, heterogeneity in ER pathway activity was indeed found [48]. Such heterogeneity is thought to be due to either evolution towards another tumor-driving signaling pathway during tumor progression, or higher metastatic capability of a subpopulation of cancer (stem) cells driven by another signaling pathway [31]. This process favors activity of other signaling pathways than the ER pathway, such as the PI3K and TGFβ pathways as drivers of metastases [36]. Overall this could be expected to have further reduced the percentage of patients with ER active metastatic tumors, and has probably negatively impacted the value of measuring ER pathway activity in the primary tumor to predict response to hormonal therapy in metastatic disease.

To investigate a prognostic role for PI3K pathway activity in primary tumors with respect to PFS in the tamoxifen treated metastatic patients, the groups of low FOXO-associated PI3K pathway activity and oxidative stress-associated PI3K pathway activity were taken together and compared with low PI3K pathway activity patients. This revealed PI3K pathway activity with or without cellular oxidative stress in the primary tumor as a predictive factor associated with shorter PFS in the metastatic patients. In addition, PI3K pathway activity was associated with poor response to tamoxifen treatment. The two clinical outcome measurements are related and PI3K pathway activity as well as cellular redox stress countered by *SOD2* may confer resistance to tamoxifen [49]. These findings are also in line with the described role of the PI3K-mTOR pathway in tumor progression and metastasis, and measurement of functional PI3K pathway activity in a tumor may provide useful information to guide treatment with the many available drugs that target this signaling pathway, e.g., PI3K and mTOR inhibitors [23].

TGFβ pathway activity was also associated with resistance to tamoxifen in our cohort, and this association has been described before for in vitro cancer model systems [50]. Indeed, oncogenic activity of the TGFβ pathway may be enabled by activity of the PI3K pathway, which is underscored by our findings [36,51,52].

Finally, the two patients with low ER but high AR pathway activity in the primary tumor showed rapid tumor progression under tamoxifen. AR staining was shown in a large clinical study to be associated with a relatively favorable outcome in ER positive patients while being a poor prognostic factor in ER negative breast cancer patients [53]. If primary tumor AR pathway activity was representative for AR activity in metastases, this may explain lack of response to tamoxifen therapy and puts these patients in the category with poor prognosis due to AR pathway activity. Even in the presence of an active ER pathway, an active AR pathway has been suggested to confer resistance to tamoxifen therapy with detrimental consequences [54,55,56]. Treatment with anti-androgen therapy may conceivably be a better choice to consider for patients with identified low ER but high AR pathway activity in the primary tumor. Further studies to investigate AR pathway activity in metastatic tumors and to explore this therapeutic opportunity are warranted.

This unique patient cohort of endocrine treatment-naïve patients with metastatic disease also has inherent limitations, since only patients who developed metastatic disease were included. Unfortunately, Affymetrix HG-U133Plus2.0 or +PM expression data were not available from a comparable group of endocrine treatment-naïve patients that did not develop metastases. Clinical confirmation would require a study in which patients, similarly treated, with and without development of metastases could be compared. Due to lack of a patient set in which patients did not receive tamoxifen treatment, it was also not possible to define the value of primary tumor ER pathway activity to predict response to tamoxifen therapy in the metastatic setting. On the other hand, clinical studies in the neoadjuvant hormonal treatment setting may serve as proper validation studies to prove the value of the ER pathway model to predict therapy response. Indeed, results of a neoadjuvant endocrine treatment study [19] provide a first confirmation that the ER pathway activity score may be useful to predict response to endocrine therapy. 

## 5. Conclusions

Based on the presented evidence, the most important conclusions of the current study are as follows. (1) Positivity for ER protein in a breast cancer sample does not always mean that the ER pathway is in the active state. (2) Despite expected heterogeneity between primary and metastatic tumors, ER pathway activity in the primary tumor retained predictive value for PFS under tamoxifen, suggesting that at least several metastases had an active ER pathway. This provides additional support for the expectation that ER pathway activity analysis has clinical utility as a predictor of hormonal therapy response in ER positive breast cancer patients in general. (3) PI3K-FOXO, HH, AR and TGFβ pathway analysis may be useful to identify patients with ER positive tumors who may benefit from alternative targeted therapeutics. Clinical studies are in progress to provide further clinical validation.

## Figures and Tables

**Figure 1 cancers-12-00802-f001:**
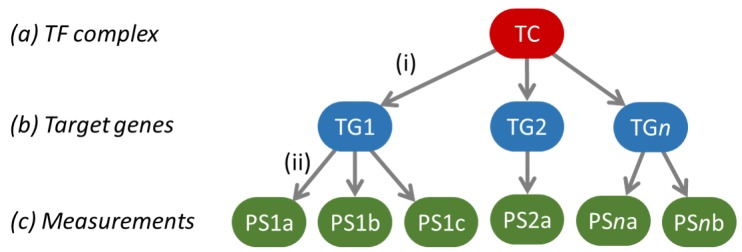
The general structure of the Bayesian network models used to model a pathway’s transcriptional program. The green ovals indicate the probesets on the Affymetrix arrays used to measure target gene levels, and the respective target genes are indicated in the blue ovals as TG 1 to n. After calibration, the model is used to infer backwards from the target gene levels what the odds are that the pathway-associated transcription factor complex (TC) is active or not, as a quantitative readout for signaling pathway activity.

**Figure 2 cancers-12-00802-f002:**
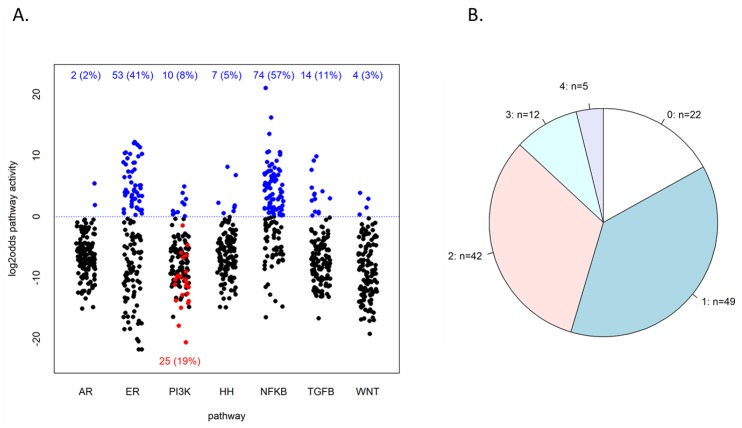
Resulting pathway activities of the study cohort. (**A**) Resulting log2odds values of pathway analysis on the 130 patient samples. The dotted blue line indicates the default threshold of 0 on the log2odds scale, and the numbers in blue at the top indicate the number of samples above this threshold. PI3K pathway log2odds score is taken as the inverse of the FOXO score, so samples below the line had a high FOXO score. Among the latter are 25 samples that also had an elevated *SOD2* expression, indicated in red. PI3K active samples can be counted as those that had either low FOXO activity or FOXO-high-*SOD2*-high (explained in the text and [4]). (**B**) Frequencies of the number of different pathways with high activity within the same sample; n = the number of samples that have 0, 1, …, 4 pathways high, as indicated.

**Figure 3 cancers-12-00802-f003:**
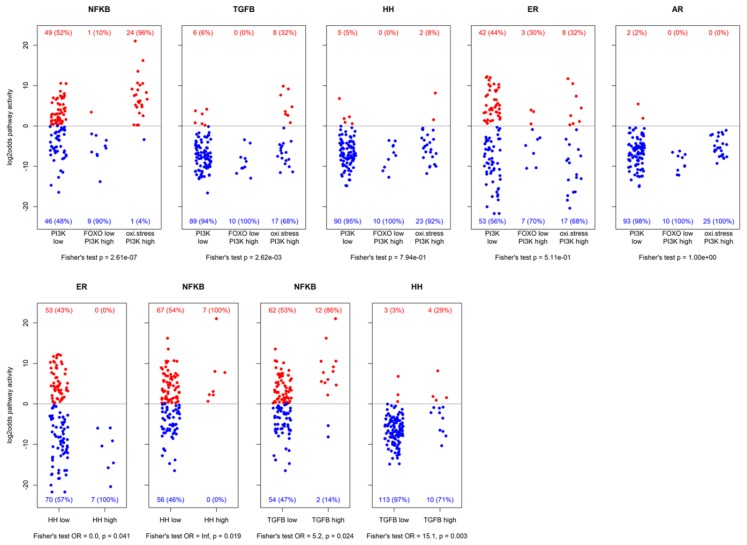
Resulting log2odds scores of pathway analysis across different groups of samples. The top row shows activity for PI3K pathway inactive, FOXO-low PI3K pathway active, and oxidative stress PI3K pathway active samples. The bottom row shows activity of some pathways vs. HH pathway activity and TGFβ pathway activity. Patient numbers are indicated with corresponding percentages (between brackets). Reported p-values are nominal.

**Figure 4 cancers-12-00802-f004:**
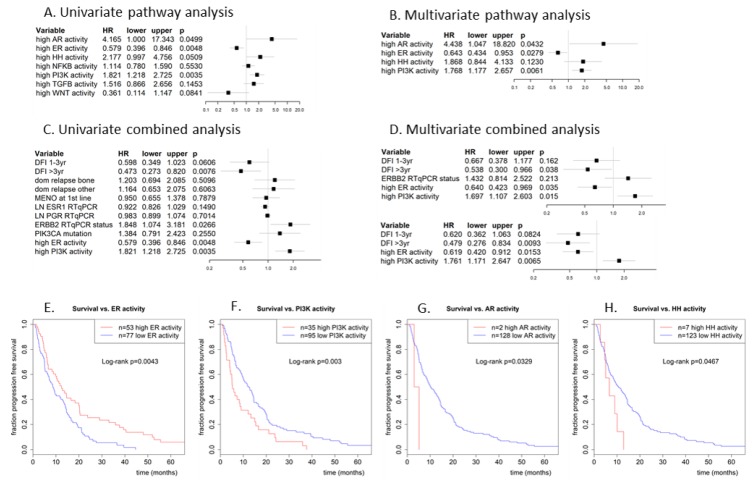
Cox proportional hazards regression analysis of progression-free survival vs. (**A**,**B**) dichotomized pathway activities (log2odds > 0) and vs. (**C**,**D**) dichotomized pathway activities combined with traditional predictors. (**A**) univariate analysis with all 7 pathways, (**B**) multivariate analysis of pathway activities that were significant in the univariate analysis, (**C**) univariate analysis of most significant pathways from (**B**) with clinical predictors, (**D**) multivariate analysis with the significant predictors from univariate analysis (**D**). Used clinical predictors are: disease-free interval (1–3yr or >3yr vs. reference DFI < 1yr), dominant site of relapse (bone or other vs. reference local), menopausal status at start 1st line therapy, log *ESR1* and *PGR* RT-qPCR expression (as continuous predictor), HER2/*ERBB2* RT-qPCR status (amplified vs. reference not amplified) and *PIK3CA* mutation status (mutated vs. reference wildype). (**E**) Kaplan-Meijer plot for progression-free survival of patients with a high ER pathway activity (log2odds > 0; red line) vs. patients with a low ER pathway activity (log2odds ≤ 0; blue line). (**F**) Similarly, for the PI3K pathway, (**G**) AR pathway, and (**H**) HH pathway. The PI3K pathway is considered to be active in case of low FOXO activity or in case of combined high FOXO activity and elevated *SOD2* expression.

**Table 1 cancers-12-00802-t001:** Patient information.

Characteristics		Count
Age at diagnosis	≤50 years	58
	>50 years	72
Menopausal status at diagnosis	pre-menopausal	50
	post-menopausal	62
	unknown	18
Grade	good/moderate	21
	poor	69
	unknown	40
pT; pathological tumor classification	≤2 cm	43
	>2 cm and ≤5 cm	71
	>5 cm + pT4	11
	unknown	5
Nodal status	no positive lymph nodes	75
	positive lymph nodes	55
Age at 1st line therapy	≤50 years	41
	>50 years	89
Menopausal status at start 1st line therapy	pre-menopausal	49
	post-menopausal	81
*ERBB2* RT-qPCR status [7]	not amplified	104
	amplified	16
	unknown	10
*PIK3CA* mutation status	wildtype	27
	mutated	27
	unknown	76
PR protein status	negative	25
	positive	96
	unknown	9
Disease free interval	<1 year	19
	1–3 years	57
	>3 years	54
Adjuvant therapy	none	106
	chemotherapy	24
Dominant site of 1st relapse	Local regional relapse (LRR)	17
	Bone	68
	Other	45

**Table 2 cancers-12-00802-t002:** Association of pathway activity in primary tumor with response to hormonal therapy in the patients with metastatic disease.

**Pathway**	**Response**	**Low ER Activity**	**High ER Activity**	**Percentage High**	**Fisher Test**
**A. ER Pathway**	CR	1	2	67%	
	PR	9	6	40%	
	SD > 6 m	38	31	45%	*p* = 0.36
	SD ≤ 6 m	5	5	50%	
	PD	24	9	27%	
	Non-PD	53	44	45%	OR = 0.45
	PD	24	9	27%	*p* = 0.10
**B. PI3K Pathway**	**Response**	**Low PI3K Activity**	**High PI3K Activity**	**Percentage High**	**Fisher Test**
	CR	3	0	0%	
	PR	12	3	20%	
	SD > 6 m	56	13	19%	*p* = 0.035
	SD ≤ 6 m	5	5	50%	
	PD	19	14	42%	
	CR, PR, SD > 6 m	71	16	18%	OR = 3.5
	SD ≤ 6 m, PD	24	19	44%	*p* = 0.003
**C. TGFβ Pathway**	**Response**	**Low TGFβ Activity**	**High TGFβ Activity**	**Percentage High**	**Fisher Test**
	CR	3	0	0%	
	PR	14	1	7%	
	SD > 6 m	64	5	7%	*p* = 0.11
	SD ≤ 6 m	10	0	0%	
	PD	25	8	24%	
	Non-PD	91	6	6%	OR = 4.8
	PD	25	8	24%	*p* = 0.008

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
