# Peer review of "ER and PI3K Pathway Activity in Primary ER Positive Breast Cancer Is Associated with Progression-Free Survival of Metastatic Patients under First-Line Tamoxifen"

_cancers, 2020, doi:10.3390/cancers12040802_

Round 1
Reviewer 1 Report
In the manuscript Sieuwerts at al. the authors analyze 130 primary breast cancer samples (from ER+ patients). while the samples were from non metastatic breast cancer, the patients developed, after surgery, metastatic disease that was treated with tamoxifen.
the authors apply a bioinformatic approach to analyze the activation status of several signaling pathways (through their target genes) and how it is associated with the clinical outcome. moreover they aim to show that ER pathway activity in the primary tumor is a better predictor of response to tamoxifen than just ER protein expression
my concerns are the following:
1) in the manuscript RT-qPCR are mentioned but there is nothing about It in the materials and method section. please provide all the necessary informations together with the raw data.
2) Lines 147-149 the authors say: "Once built, the models were biologically validated on human cell line models for the respective signal transduction pathways and on tissue samples of cancer types that are well characterized with respect to activity of the specific signal transduction pathway". while they provide references I think it important to explain better how this validation has been performed and in particular which cell lines were used.
3) a table containing age at diagnosis, grade and histology for all the patients should be avalaible
Author Response
1) in the manuscript RT-qPCR are mentioned but there is nothing about It in the materials and method section. please provide all the necessary informations together with the raw data.
We added text and references on this in the section, and provided data per patient in a supplementary table.
2) Lines 147-149 the authors say: "Once built, the models were biologically validated on human cell line models for the respective signal transduction pathways and on tissue samples of cancer types that are well characterized with respect to activity of the specific signal transduction pathway". while they provide references I think it important to explain better how this validation has been performed and in particular which cell lines were used.
We summarized the cell lines and other kinds of samples that have been used for this. Including all details is beyond the scope of the current manuscript, in our view, so for those we refer to the undelying papers.
3) a table containing age at diagnosis, grade and histology for all the patients should be available.
This has been added to the supplementary table mentioned under 1).
Reviewer 2 Report
Sieuwerts et al have analysed 130 samples from non-metastatic ER+ breast cancer patients treated with surgery without adjuvant hormonal therapy, who subsequently developed metastatic disease using mRNA expression data. They then used novel computational models too determine if the molecular phenotype of a cancer is predictive of therapy response.
This is a well written and interesting paper, and from the perspective of a non-computational biologist was easy to understand.
I have a few minor comments/ queries.
- The authors say that tissue samples from the primary tumour contained at least 30% invasive tumour cells. Does this refer to an FFPE tumour sample or to the FF tumour that was used for the analysis. Please clarify in the methods and if referring to the FF tumour please clarify how the % tumour was determined.
- Was the mutation status of these tumours known? For example can PI3K pathway activity be explained by mutations in PIK3CA or other PI3K pathway genes.
- If PIK3CA/PI3K pathway mutation status is known it should be included in the multivariate analysis
- Minor grammar mistakes throughout the manuscript should be addressed
Author Response
1) The authors say that tissue samples from the primary tumour contained at least 30% invasive tumour cells. Does this refer to an FFPE tumour sample or to the FF tumour that was used for the analysis. Please clarify in the methods and if referring to the FF tumour please clarify how the % tumour was determined.
We added a sentence that this was done on FF material, and refer to a previous paper for details.
2) Was the mutation status of these tumours known? For example can PI3K pathway activity be explained by mutations in PIK3CA or other PI3K pathway genes.
An earlier publication showed that PIK3CA mutation status was not associated with outcome on tamoxifen treatment. Nevertheless, mutation information was available for 54 of the 130 samples, and we added the analysis against PI3K pathway activity and progression free survival. No significant correlations were found, though.
3) If PIK3CA/PI3K pathway mutation status is known it should be included in the multivariate analysis.
See above response.
4) Minor grammar mistakes throughout the manuscript should be addressed.
We carefully re-read the manuscript to address these.
Reviewer 3 Report
Authors have evaluated the effect of different signaling pathways activity on progression-free survival time of Tam treated ER+ patients and observed that high PI3K and TGFβ pathway activities lead to worse response toward treatment. PI3K plays a very important role in the regulation of mTOR pathways which impacts different important aspects of cancer cell physiology. It will be interesting if the author can evaluate the link between Pi3K-mTOR and its impact on PFS. Overall, manuscripts finding are significant and will help to improve our understanding of the differential response of breast cancer towards tamoxifen.
Author Response
Authors have evaluated the effect of different signaling pathways activity on progression-free survival time of Tam treated ER+ patients and observed that high PI3K and TGFβ pathway activities lead to worse response toward treatment. PI3K plays a very important role in the regulation of mTOR pathways which impacts different important aspects of cancer cell physiology. It will be interesting if the author can evaluate the link between Pi3K-mTOR and its impact on PFS. Overall, manuscripts finding are significant and will help to improve our understanding of the differential response of breast cancer towards tamoxifen.
We agree with the referee, and indeed activation of the PI3K pathway generally results in activation of mTOR, and mTOR also plays a role in progression and metastasis, and can be targeted by specific drugs, like everolimus, as we described before [van de Stolpe, Cancers 2019]. We have added a sentence on this in the discussion. Unfortunately, we do not have a computational model of the mTOR pathway, so we could not evaluate this further.